# Impacts drive lunar rockfalls over billions of years

Valentin Tertius Bickel [1,2✉], Jordan Aaron [2], Andrea Manconi [2], Simon Loew [2] & Urs Mall [1]

Past exploration missions have revealed that the lunar topography is eroded through mass wasting processes such as rockfalls and other types of landslides, similar to Earth. We have analyzed an archive of more than 2 million high-resolution images using an AI and big data-driven approach and created the first global map of 136.610 lunar rockfall events. Using this map, we show that mass wasting is primarily driven by impacts and impact-induced fracture networks. We further identify a large number of currently unknown rockfall clusters, potentially revealing regions of recent seismic activity. Our observations show that the oldest, pre-Nectarian topography still hosts rockfalls, indicating that its erosion has been active throughout the late Copernican age and likely continues today. Our findings have important implications for the estimation of the Moon's erosional state and other airless bodies as well as for the understanding of the topographic evolution of planetary surfaces in general.

[1] Department Planets and Comets, Max Planck Institute for Solar System Research, Goettingen, Germany. [2] Engineering Geology, Department for Earth Sciences, ETH Zurich, Zurich, Switzerland. ✉email: bickel@mps.mpg.de

Over geologic timescales, dynamic erosion processes smooth topography and shape planetary surfaces[1,2]. On airless bodies, such as the Moon, erosion has primarily been thought to occur through space weathering[3]. However, recently acquired, high-resolution imagery has revealed a startling feature on lunar slopes: ubiquitous mass-wasting features. These include granular flows[4,5], slides, slumps, and creeps[1,6], as well as rockfalls[1,7–9], a process where boulders are released or ejected from topographic highs and fall, roll, bounce, and slide to topographic lows. Rockfalls carve tracks into the lunar surface, which provide a record of the dynamic emplacement process. The characteristic combination of a displaced boulder or rock fragment and its track—here called rockfall—enables unambiguous detection from satellite imagery (see inset in Fig. 1). During the Apollo 17 mission, astronauts Jack Schmitt and Eugene Cernan directly sampled a rockfall at station 6 in Taurus Littrow valley and returned multiple samples to Earth. Using these data in combination with orbital imagery, rockfall track survivability has been estimated to range from ~1.55 to 35 Ma[2,8–12] with an upper limit of ~150–300 Ma[13], meaning that the tracks observed today are the expression of geologically recent (late Copernican)—and potentially current—erosion processes.

Past studies have mapped lunar rockfall features for selected regions across the lunar equatorial and polar highland, mare, pyroclastic, and permanently shadowed regions[7–9,14,15]. These studies have hypothesized that the main drivers of lunar rockfalls are shallow and deep moonquakes, impact-induced shaking, and thermal fatigue caused by the extreme temperature variations on the Moon[7,16]. Here we use the term rockfall driver to represent both the long-term cause (also called precondition) and short-term trigger, as observations in satellite imagery only allow for an explicit distinction of rockfall triggers and causes in some cases[8,9]. Other studies have suggested that major basin-forming impact events can cause intense and global seismic shaking that results in erosion of steep slopes through large-scale mass wasting[17,18]. As the latest basin-forming event, the Orientale impact, occurred ~3.8 Ga ago, all topography older than ~3.8 Ga would feature shallow slope angles and, thus fewer mass-wasting processes[18], a finding supported by an apparent absence of mass-wasting features like rockfalls and granular flows on the oldest—(pre-)Nectarian—topography[7]. Thus the current assumption is that these ancient, (pre-)Nectarian terranes represent the final evolutionary stage of the lunar surface as they have shallow slopes[18,19] and may not host mass-wasting features. However, these past studies have focused on small, isolated regions[8,9] and/or applied coarse spatial sampling[7].

A high-resolution, global map of mass-wasting features such as rockfalls is required to identify and better understand their spatial distribution and drivers, as well as the state of lunar erosion activity. We have analyzed an archive of >2 million high-resolution images taken by the National Aeronautics and Space Administration's (NASA's) Lunar Reconnaissance Orbiter (LRO)

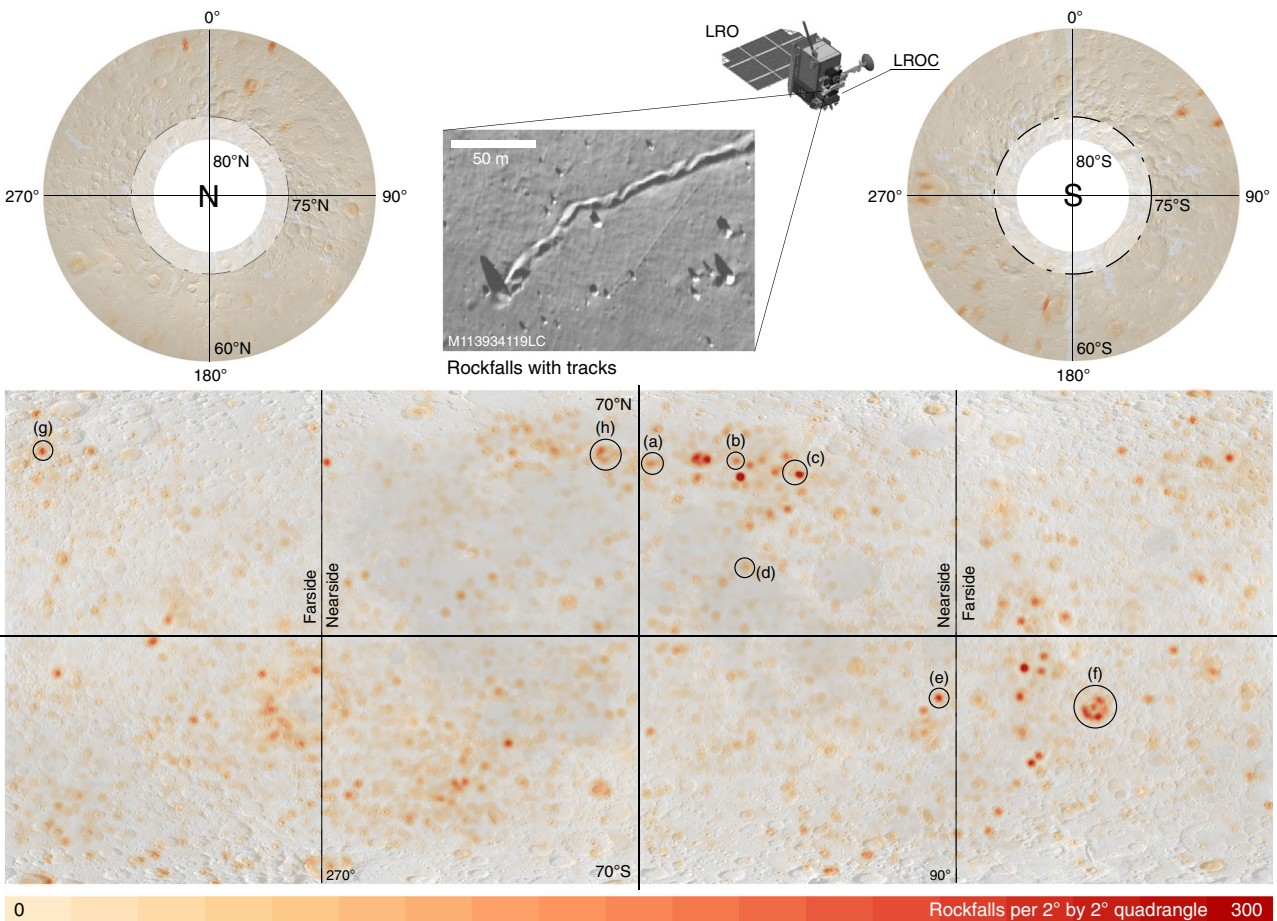

**Fig. 1 Global rockfall distribution heat map.** Heat maps of lunar global rockfall distribution in both equirectangular (70°N to 70°S) and polar projection (60°N to 80°N and 60°S to 80°S). Mapping results above 75°N and below 75°S might be less complete due to challenging illumination conditions (indicated by transparent white areas). Rockfall density is given in rockfalls per square degrees, i.e., the number of rockfalls per 2° latitude by 2° longitude quadrangle. The labeled black circles denote details shown in Fig. 2. Inset shows examples of rockfalls and their tracks with various sizes: detail of LRO NAC (LROC) image M113934119LC. Global WAC mosaic in the background[40]. NAC image credits to LROC/ASU/NASA.

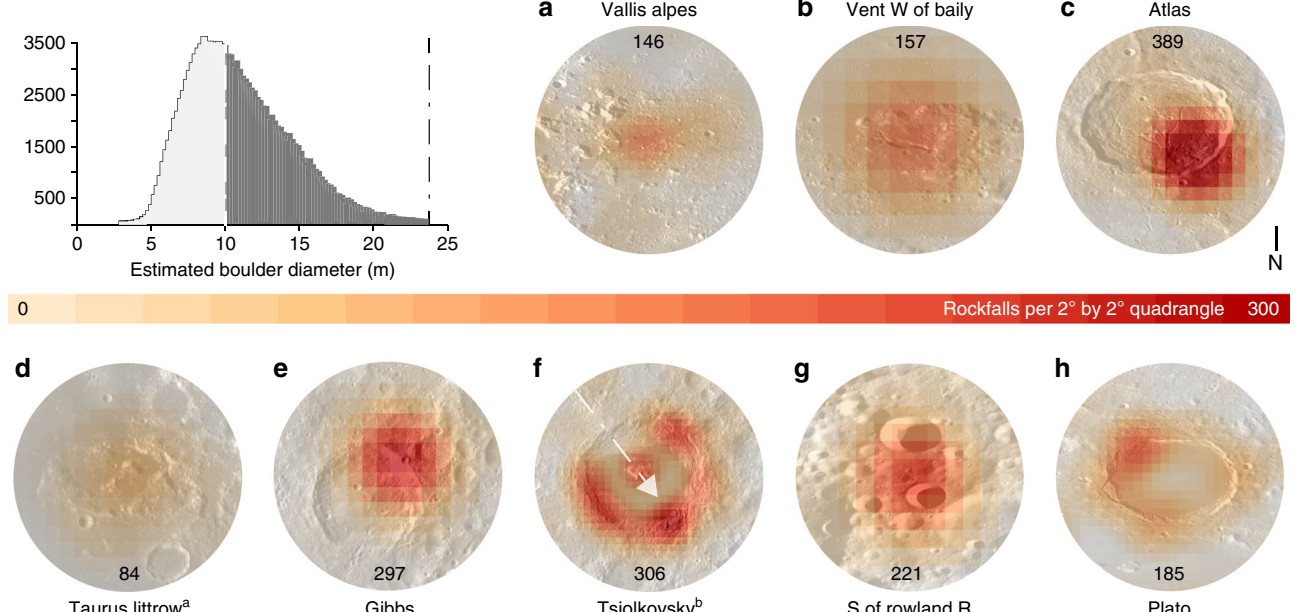

**Fig. 2 General rockfall diameter distribution and details of Fig. 1.** The histogram shows the general distribution of estimated rockfall diameters. Estimated rockfall diameters are based on the output of the neural network and are potentially biased for smaller diameters (section indicated in white), as the CNN's detection accuracy is reduced when detecting small features. Insets **a–h** are details from Fig. 1 and represent graben (**a**), pyroclastic vents (**b**), craters (**c, f, h**), regular slopes (**d**), and recent impacts (**e, g**). Some craters show a heterogeneous distribution of rockfall abundance, such as **c, f, h**. The peak count of spatial rockfall density is shown in the insets per 2° by 2° quadrangle. In **e, g**, we identified a series of rockfalls that are directly induced by small-scale impact events (impact-ejected); for details refer to Supplementary Fig. 2. [a]Apollo 17 landing site, [b]Morse et al.[30] proposed an impact trajectory for **f** based on the distribution of the ejecta blanket (axis NW–SE), which agrees with the rockfall distribution (white arrow). Global WAC mosaic in the background[40].

narrow angle camera (NAC) using a convolutional neural network (CNN)[20] in combination with advanced cloud computing capabilities and created the first global map of the location and size of lunar rockfalls.

## Results

**Spatial distribution of lunar rockfalls.** We identified 136,610 rockfalls across the lunar surface in the latitude range from 80°N to 80°S, with a mean spatial density of ~2 rockfalls per 1° by 1° quadrangle. The spatial distribution of rockfalls is heterogeneous, with the majority of features located in highland terranes (91%). The distributions over the lunar near side and far side, as well as the northern and southern hemispheres, are relatively well balanced. There are distinct and pronounced clusters of rockfalls across all terranes (Fig. 1), while the local spatial distribution of rockfalls within selected craters is asymmetric, as is particularly obvious in Tsiolkovsky (Fig. 2f), Atlas (Fig. 2c), and Plato craters (Fig. 2h). The peak rockfall density of 272 rockfalls per 1° by 1° quadrangle has been measured in the Bürg crater on the lunar nearside. Other remarkable, high-spatial density rockfall clusters are located in Mare Orientale basin and the associated Montes Rook and Montes Cordillera, as well as in Tsiolkovsky, Pasteur D, Atlas, Aristoteles, Milne N/L, and Crookes craters. The heterogeneous spatial distribution of rockfall clusters could potentially reveal regions of recent seismic activity. Lunar rockfall (boulder) diameters range from ~3 to ~25 m, while the majority of rockfall (boulder) diameters range from ~7 to ~10 m (Fig. 2). The ubiquitous and homogeneous soil on lunar slopes[21] would act in favor of larger deposited boulder diameters, since a large portion of the kinetic energy would be absorbed by the soil and would thus not contribute to boulder fragmentation. On average, older lunar regions appear to host boulders with systematically larger volumes (Supplementary Fig. 1). A potential explanation of this observation could be an increase of the lateral spacing between

impact-induced radial fractures as the crater erodes over time, producing larger boulders.

**Impacts are the main driver of lunar rockfalls.** We found evidence that impact processes directly trigger lunar mass wasting (Fig. 2e, g) and, given the likely age of rockfall tracks, actively drive the erosion of the lunar topography (Supplementary Fig. 2). One variation of these impact-ejected boulders features tracks that spread radially away from the impact site (Fig. 2g), where the gradually decreasing distance between individual jumps of each boulder suggests that the boulders were initially airborne and subsequently lost more and more energy along their track. A second variation features an impactor that hits the slope of an existing, larger crater (Fig. 2e), resulting in instant displacement of a number of boulders across the slope of the older host crater. However, the majority of rockfalls occur on crater slopes without the direct influence of a subsequent impactor. This indicates that impacts act not only as a direct trigger but also induce fracture networks that represent the main preconditioning factor for lunar rockfalls. Subsequently, other, more recent geological processes, such as tectonic and/or volcanic activity, can potentially act as alternative or additional triggers of these impact-fractured rock masses.

The majority of rockfalls are located in impact craters or basins (~84%), while the remaining features are situated in graben or along faults and scarps (~0.8%), on volcanic edifices or in vents and rilles (~0.4%), and regular slopes, which include areas with unclassified geomorphic contexts (~14.8%) (Supplementary Fig. 3). If craters with diameters between 1 and 5 km are neglected[22], the portion of rockfalls located in craters or basins is ~72.7% and ~26.1%, respectively, on regular slopes, including unclassified geomorphic contexts. The percentage of rockfalls in craters is potentially still underestimated (and unclassified overestimated), as craters with diameters <1 km have not yet been mapped on a global scale. Most mapped rockfalls are located in craters with

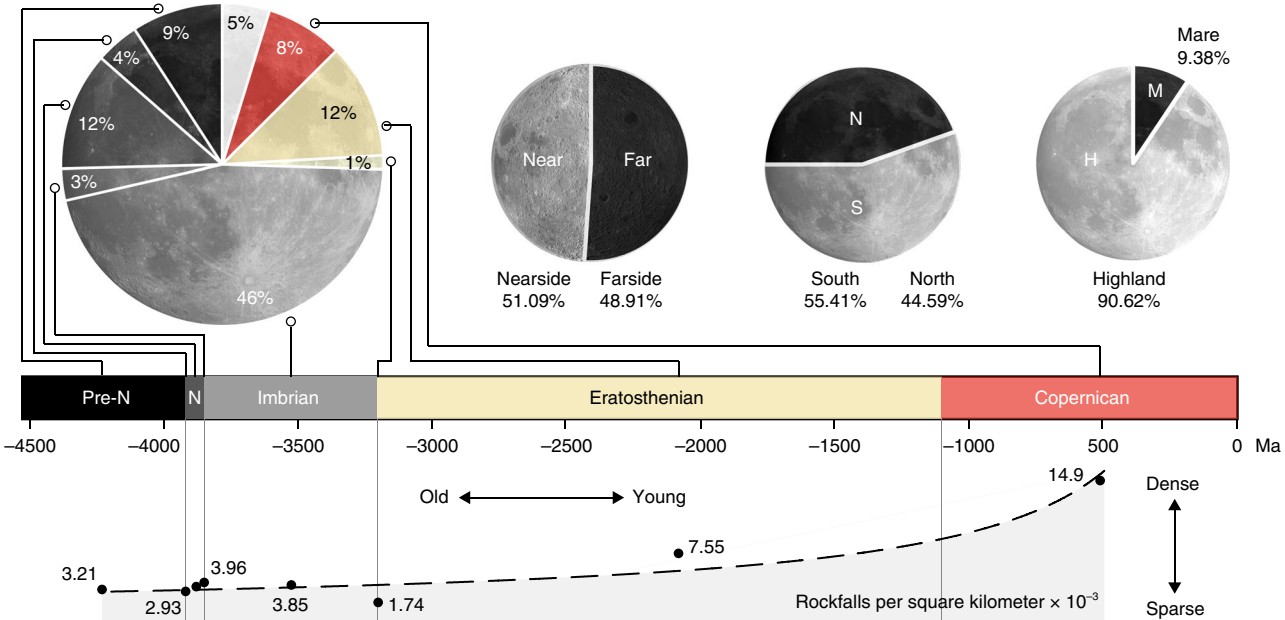

**Fig. 3 Statistics about rockfall distribution in space and time.** Statistics highlight the spatial distribution of rockfalls and its implications. More rockfalls occur in highland terranes and the majority of rockfalls occur in Imbrian terranes, while Nectarian (N) and even pre-Nectarian sites (pre-N) feature a substantial number of recent rockfalls (~25%). Owing to inaccuracies in the underlying geologic map, the number of rockfalls in ancient terranes is potentially reduced in favor of the Copernican and Eratosthenian bins (see Supplementary Fig. 5). The normalization of the CNN-derived rockfall counts over area in the regions provides a first-order estimate (power law fit, black dashed line) of the advective erosion rate of the lunar topography and its evolution over time (Eq. (1) in the "Methods" section). Geologic lunar timescale based on refs. [56,57].

diameters <25 km and the spatial density of rockfalls in these smaller craters is on average three times higher than in the other analyzed geomorphic regions, i.e., $\sim 13E^{-3}$ vs $\sim 4E^{-3}$ rockfalls per square kilometer. We do not observe a general increase in rockfall number or spatial density within or in the direct vicinity of tectonic features, such as graben and faults, which would be expected if recent tectonic activity is a main driver of lunar rockfall. Two potential explanations of this observation are: (1) endogenic seismic activity is a relatively unimportant trigger or (2) there is a lack of impact-induced fracture networks (precondition) for high-frequency rockfall occurrence in tectonic geomorphic contexts. At present, current data cannot distinguish between these two hypotheses. There are limited exceptions to this observation, such as the rockfall clusters in Vallis Alpes (a graben, Fig. 2a and Supplementary Fig. 3c), Aristarchus plateau (a volcanic region, Supplementary Fig. 3a), and a pyroclastic vent west of the Baily crater (Fig. 2b). Thus, while rockfall frequency and density peak in impact craters, particularly in young and steep[18] craters (e.g. Fig. 3), i.e., lunar rockfalls appear to be predominantly driven by impacts, our results show that recent tectonic and potentially volcanic activity contribute as well. This means that our global map of rockfalls could also be applied to better understand, localize, and constrain the current tectonic—and potentially volcanic—activity of the Moon.

**Rockfalls and the erosional state of the lunar surface.** The majority of rockfalls occurs in Imbrian terranes (~46%; Fig. 3), but a substantial number of rockfalls (~25%) are located in ancient Nectarian and pre-Nectarian terranes and craters with estimated ages of >3.8 Ga (Fig. 3; Supplementary Fig. 4). This is a remarkable observation, as the current assumption is that these ancient terranes only feature shallow slopes[18] and would not host any mass-wasting features other than slope creep[7]. The mean spatial density of rockfalls as a function of geologic epoch peaks in very young (Copernican) craters and declines over time

(Fig. 3). This peak is expected, as fresh topography is usually sharp and erosion rates are high. As erosion proceeds, the topography becomes smoother[18,19] and the rockfall density declines. We find that after ~1.5 Ga the density of rockfalls is reduced by roughly 50% compared to the (initial) Copernican density (see Fig. 3). This observation implies that the spatial density of rockfalls could be used to complement the analysis of the age of lunar topography (see Eq. (1) in the "Methods" section) as derived from diffusive erosional processes[19]. We note that inaccuracies in the geologic map may influence these counts (Supplementary Fig. 5); however, given the global coverage of our rockfall map (Fig. 1) and manual checking of detections (Supplementary Fig. 4), we think that mapping inaccuracies cannot entirely account for the presence of rockfall in ancient terranes.

In contrast to the counting of rockfalls in terranes of specific ages, the estimate of the time of occurrence of rockfall events is controlled by the survival time of the tracks carved by the displaced boulders. If boulder tracks erode quickly, the mapped rockfall features would indicate very recent erosional activity; if tracks survive for a long time, the occurrence of rockfalls is more difficult to constrain. Survival times of boulder tracks have been estimated to lie between ~1.55 and 35 Ma using observations made by astronauts, dating of in situ samples, and satellite imagery[2,8–12]. However, track survival times might vary at different locations due to variations in impactor influx and local topographic differences[10]. A maximum survival time of tracks between ~150 and 300 Ma can be established by using the mean survival time of lunar boulders >2 m[13], assuming that boulders would outlast their tracks. Thus our observations of rockfalls in (pre-)Nectarian terranes (>3.8 Ga old) imply that these ancient terranes have been actively eroded in the past 300–~1.55 Ma and likely are subject to erosion today. One direct consequence of this observation is that ~3.8 Ga are not sufficient to completely erode the relief of an airless and dry body, even if slopes underwent instantaneous, global, impact-induced large-scale mass wasting such as hypothesized through, e.g., the Orientale impact ~3.8 Ga

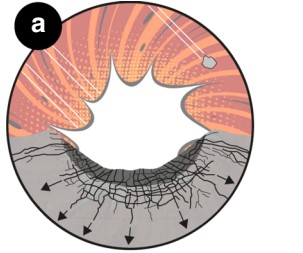 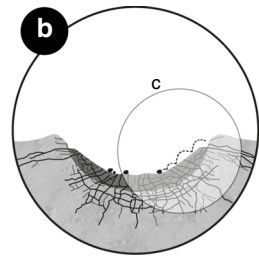 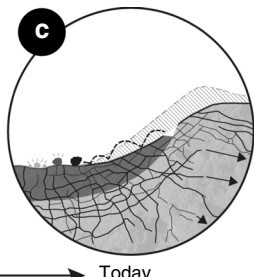

*t* > 3.8 Ga Today

**Fig. 4 Conceptual model of lunar mass wasting. a** Impacting bodies batter the lunar surface, creating topography (craters and associated terrain) as well as intense fracturing of the bedrock. **b** Impact craters feature radial, concentric, shear, and spalling fractures that cause the release of boulders—rockfalls—over time. **c** The extent of fracture networks is sufficient to drive rockfalls over billions of years. Fracture distribution pattern taken from ref. [27].

ago[17,18]. This observation also indicates that the oldest surfaces of other bodies in the solar system with similar properties and impact histories as the Moon might still be subject to (small-scale) erosion as well, such as the pre-Calorian inter-crater plains on planet Mercury[23,24] or the pre-Veneneia crust on asteroid 4Vesta[25].

## Discussion

The results presented here substantiate that impacts are the primary driver of lunar rockfalls. A substantial proportion of the observed lunar rockfalls occur in craters >3.8 Ga, suggesting that the legacy of impacts as a driver of rockfall activity lasts for geologic timescales. Previous work has shown that impacts cause substantial fracturing of the target bedrock, including sets of radial, concentric, spall, and shear discontinuities[26–28]. On Earth, discontinuities are the main preconditioning factor for rockfalls[28,29], and our results indicate that the same is true for the Moon (Fig. 4). Additional evidence for impact-induced fracture systems governing rockfall occurrence is provided by the spatially asymmetric occurrence of rockfalls in craters such as Tsiolkovsky on the lunar far side, which agrees with the inferred impact direction(-axis) along a NW–SE trajectory[30] (Fig. 2f and Supplementary Fig. 3d). An oblique impact angle would cause a denser and more consistent fracture network on the far side of the crater that predisposes the production of an increased number of rockfalls over time. In addition, the morphology and density of the impact-induced fracture systems controls the magnitude of lunar rockfalls that are considerably larger than on Earth, on average[31].

The unexpected and long-lasting effect of pre-Nectarian impactors on the—potentially ongoing—erosion of ancient terranes could be favored by an increased impactor influx during the early days of the Earth–Moon system. It has been hypothesized that there was a systematically higher flux of impactors during the (pre-)Nectarian period—sometimes referred to as the Late Heavy Bombardment—that transitioned to an impactor population with lower impact ratios roughly 3.5–3.8 Ga ago[32,33] or potentially even earlier in the Moon's history[34]. A significantly increased number of impacts could potentially result in a more pronounced fracture network and loss of rock bridges in (pre-)Nectarian rock masses that drive mass wasting over a period of >4 billion years.

## Methods

**Deep learning-driven rockfall detection and mapping.** We applied a CNN to automatically search for lunar rockfalls in the image archive of NASA's LRO NAC, which contains >2 million entries. The used CNN is the fifth and most recent generation (M5) of a series of networks that have been trained, validated, and tested in a previous step[20]. M5 has been trained with 809,550 augmented rockfall images using an implementation of a state-of-the-art single-stage dense object detector (RetinaNet[35,36]) in combination with a ResNet101 backbone[37] and using open-source-only software (Python, Keras, TensorFlow). In the testing set, M5 achieved an average precision score of 0.89, a recall of 0.44–0.69, and a precision of 0.98–1.00 for confidence levels of 50% and 60%, respectively[20]. A confidence level of 60% was selected for the analysis as it provides the optimal compromise between

recall and precision, based on the testing set; it has to be noted that M5's performance on the global dataset might deviate from the performance in the testing set. M5 was applied to optical, single-band, high-resolution satellite imagery that is acquired by NASA's LRO, a lunar spacecraft that has been in polar orbit around the Moon since 2009[38]. The spatial coverage by LRO NAC is excellent and almost global, with limitations of coverage in the proximity of the lunar poles, due to the challenging illumination conditions and shadows.

We used pre-calibrated and compressed NAC pyramid image files as they have the maximal spatial resolution but a reduced bit depth (8 bit) and file size, allowing for optimized processing speeds. We accessed these data through a customized NASA JPL Moon Trek api (application programming interface) using an image selection algorithm that selected NAC images based on three criteria: (1) spatial resolution, (2) solar incidence angle, and (3) overlap with surrounding images. This routine uses the Klee algorithm[39] to first calculate the overlap of neighboring NAC images and then find the best combination of images that cover the area of interest with maximum underlap and minimum overlap. Overlap needs to be minimized in order to avoid processing of redundant imagery and to avoid detections of physically identical rockfalls in different images. Underlap needs to be minimized to avoid gaps in the coverage and to achieve a seamless scanning of the lunar surface. The algorithm prioritizes images with the highest spatial resolution and with incidence angles between 30 and 60°, wherever possible. After identification of the NAC images, the data are tiled and input into the CNN. Subsequently, all detections are extracted and their positions reconstructed in real-world coordinates using the associated meta data from the LROC image archive (see Supplementary Fig. 6). In addition, the size of each detection's bounding box (predicted by the CNN) has been used to estimate the size of the rockfall[20].

We used a total of 2 local computers and 9 Google Cloud instances (AI Notebooks) to process the data, with a total of 30 central processing units and 15 graphics processing units (NVIDIA Titan Xp, NVIDIA GTX 1080Ti, NVIDIA Tesla K80, NVIDIA Tesla V100). The access to the cloud instances was provided by the NASA Frontier Development Lab, its numerous sponsors, and a Google Cloud Academic Research Grant. We processed a total of 240,401 NAC images to achieve global coverage, totaling to ~64 Tpixels and ~7.5 TB of data.

**Limitations of the deep learning-driven approach.** The limitations of CNN-driven rockfall detection and mapping have been extensively discussed in previous work[20]. The main limitations are: (1) spatial gaps between NAC images due to incomplete coverage or poor image selection, (2) unfavorable lighting conditions (shadows), (3) bad images (noise-only), and (4) poor performance of the CNN (mis-classifications or missed detections). All possible steps were taken to minimize the influence of these limitations, such as the implementation of a routine that discards bad NAC images and the minimization of mis-classifications (false positives) by selecting a CNN confidence level that achieves high detection precisions while maximizing recall. With the selected 60% confidence level, M5 achieves an expected recall between 0.44 and 0.69 and a precision of around 0.98, meaning that a portion of rockfalls is not detected (false negatives) and that there is a limited potential for false detections to occur (false positives)[20].

**Global analysis of rockfall distribution.** The results from the global CNN-driven rockfall mapping were analyzed in combination with other relevant data and maps in an open-source Geographic Information System (QGIS). Auxiliary data included a global LRO wide angle camera mosaic[40], a Digital Terrain Model[41], and a geologic map by the United States Geologic Survey[42,43]—version May 2019. Other data were retrieved from the LROC vector data archive, such as shape file data for highland–mare boundaries[44], craters between 5 and 20 km in diameter[45], the locations of Copernican craters[46], and lobate scarps[44]. Additional data was added to QGIS, specifically information about graben and other tectonic features[47–49], volcanic edifices and features[50], craters between 1 and 5 km[22], and craters >20 km in diameter[51].

Volcanic edifices on the Moon are predominantly regional topographic swells called shield volcanoes[50]. These edifices have diameters of tens to hundreds of kilometers and elevations of several thousand meters, while their slopes are gentle

with average angles of about 1°[50]. Volcanic domes are similar in size to shield volcanoes but feature steeper slopes of up to 20°[52]. Other volcanic features, such as sinuous rilles and pyroclastic vents, can feature even steeper slopes of up to ~25°, with depths and widths of hundreds of meters[53]. Tectonic features on the Moon with a topographic expression are, e.g., graben, wrinkle ridges, scarps, and faults with slope angles around ≥15°[54] and lengths between hundreds to several thousands of meters[49]. Craters are the dominant topographic feature on the lunar surface, where the steepest slopes of >45° are usually associated with the youngest craters. Over time, the slopes of craters erode and degrade, resulting in slope angles <~35° for pre-Imbrian craters, >3.8 Ga[18].

Some of the auxiliary data used were not directly available as vector data and had to be digitized first (e.g., information from refs. [48–50]). In addition, some of the auxiliary data are incomplete or scarce, as they were not intended to provide information on a global scale. However, no other data are currently available, which means that the results of the geomorphic rockfall distribution analysis are controlled and potentially affected by the amount and quality of the available auxiliary data. Wherever possible and feasible, the imported information was updated and optimized. It should be noted that some of the used auxiliary data have associated uncertainties and, potentially, errors, such as, for example, the crater map with diameters between 1 and 5 km (diameter)[22]. The mapping of other features, such as craters <1 km, is beyond the scope of this work and has not been performed. As more accurate, consistent, and global auxiliary data become available, the results of this study can be updated and improved.

We worked with the Moon2000 reference system in a (1) equirectangular, (2) north polar-stereographic, or (3) south polar-stereographic projection system, depending on the type of analysis and the type and properties of the used auxiliary data. As geomorphic features tend to overlap, e.g., small craters tend to be located in larger craters, all feature polygons were dissolved, i.e., all overlapping polygons were merged, in order to avoid double counts in overlapping polygons. The geologic map[42,43] was refined by removing a number of overlapping polygons in the region around 50°W and 50°E (50°N to 50°S) and merging the six individual maps to create a seamless, consistent, and global map. For the analysis, the global geologic map[42,43] was used in (1) an equirectangular (45°N to 45°S, maps I-0703, I-0948, I-1034, and I-1047), (2) a north polar-stereographic (45°N to 80°N, map I-1062), or (3) a south polar-stereographic projection (45°S to 80°S, map I-1162). Owing to the challenging polar illumination conditions, we expect a reduced number of detections in the region poleward from ~75°N to ~75°S, but we included latitudes up to 80°N and 80°S in the analysis to preserve its completeness. Rockfalls were counted in eight temporal classes (epochs): Copernican, Eratosthenian, Eratosthenian–Imbrian, Imbrian, Imbrian–Nectarian, Nectarian, Nectarian–pre-Nectarian, and pre-Nectarian. Owing to inconsistencies in the geologic map, a small number of classes needed to be discarded, e.g., a null class as well as the pre-Imbrian class, as it could not be classified as Imbrian–Nectarian, Nectarian, Nectarian–pre-Nectarian, or pre-Nectarian.

The spatial density of the mapped rockfall features has been calculated in two different ways: (1) per 1° by 1° or 2° by 2° quadrangle and (2) per square kilometer. Metric 1 has been applied for the global spatial density analysis and visualization, as the counts per quadrangle (ranging from 0 to >300 rockfalls) are easier to comprehend than the counts per square kilometer (~4.0E−03–12.9E−03 rockfalls/km$^2$). In addition, a spatial resolution of one square kilometer would result in a small spatial representation of rockfall clusters in the global map. Metric 2 has been applied to express the normalization of the rockfall counts over their host area (geomorphic or geologic), as these are usually provided in square kilometer. In either case, the spatial densities are calculated considering the projection system of the GIS.

**Estimation of rockfall diameter.** The used CNN predicts a rectangular bounding box for each detected rockfall (see Supplementary Fig. 6). Based on the diameter of a box and the spatial resolution of the used image, we reconstructed an approximation of the boulder's diameter[20]. The used transformation function has been calibrated using several landmark boulders in combination with detections of these landmark boulders in different images with a large variety of spatial resolutions, ranging from ~0.4 to ~1.5 m/pixel.

**Estimation of rockfall driven erosion rates.** We use the rockfall counts per geologic epoch (eight bins, as described above) to qualitatively assess their erosional state, where more rockfalls indicate a higher advective erosion rate and fewer rockfalls indicate a lower advective erosion rate. The count for each epoch has been plotted in the temporal center of its respective epoch as well as at the borders between epochs (see Fig. 3). We applied a power law fit to the plotted data and found a quantitative description of the approximated age and erosional state of lunar topography based on the spatial density of rockfalls, described with:

$$\gamma_{\text{Rockfall}} = 2.6358 a^{-0.813} \tag{1}$$

with $a$ as the age in Ma and $\gamma_{\text{Rockfall}}$ as rockfalls per square kilometer (the spatial rockfall density) (Fig. 4). It has to be noted that Eq. (1) represents a first-order approximation, as the fit will change if the underlying timescale changes and as it depends on the quality of the used geologic map (for the rockfall counts, see Supplementary Fig. 5). The fit also depends on the performance of the used CNN, i.e., its recall and precision: Eq. (1) should only be applied in combination with rockfall counts made by CNN M5 (used for this study)—a human will perform

differently, potentially biasing an age estimate derived with Eq. (1). The normalization of Eq. (1) with the Copernican spatial rockfall density results in a model-independent equation:

$$\% \text{ rockfall relative to Copernican} = \frac{2.6358 \, a^{-0.813}}{14.9 e^{-3}} \tag{2}$$

where at an age of 500 Ma the relative density is ~1, at 1.5 Ga ~0.5, and at 3.2 Ga ~0.25. Interestingly, our quantitative best fit agrees with the qualitative conclusion of an Apollo-era study which suggested that lunar mass wasting tempers off over time and that this trend could be described with a logarithmic fit, based on the assessment of Apollo 10 imagery[55]. The measure of spatial rockfall density could be combined with other methods to estimate crater age, such as topography diffusion[19].

## Data availability

The used image data are freely available at http://wms.lroc.asu.edu/lroc/search and the auxiliary data can be downloaded from https://www.usgs.gov/centers/astrogeology-science-center/data-tools and http://wms.lroc.asu.edu/lroc/rdr_product_select. The list of rockfall detections that support the plots within this paper and other findings of this study are available from the corresponding author upon reasonable request and from ETH Zurich's Research Data repository after a PhD thesis-related embargo period of 12 months.

## Code availability

The underlying neural network architecture RetinaNet can be accessed here: https://github.com/fizyr/keras-retinanet. To facilitate access, a newly developed web tool with a graphical user interface (using the developed neural network) will soon be available on NASA JPL's Moon Trek webpage that can be accessed here: https://trek.nasa.gov. Other auxiliary code is available from the corresponding author upon reasonable request.

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

## Acknowledgements

The authors would like to thank Emily Law, Natalie Gallegos, and the rest of the Solar System Treks team for providing access to LRO's NAC image archive via JPL's Moon Trek api and for their general support. The authors gratefully acknowledge the support by the NASA Frontier Development Lab—and its numerous sponsors—with computational resources on the Google Cloud Platform. This work has been supported with a Google Cloud Academic Research grant and an NVIDIA Academic GPU grant. One of the authors (V.T.B.) further acknowledges the financial support by the Engineering Geology group, Department of Earth Sciences, ETH Zurich, and the International Max Planck Research School (IMPRS) at the Max Planck Institute for Solar System Research (MPS). Parts of the NAC image selection algorithm have been developed as part of the SpaceHack Vol.1 in Göttingen that has been held in partnership with MPS, NASA JPL, SerNet, and GWG in May 2019.

## Author contributions

V.T.B, A.M., S.L., and U.M. planned the project. V.T.B. developed, implemented, and optimized the used convolutional neural network and processing routines and initiated and used the local and cloud platform resources to process the data. V.T.B. performed the main part of the analysis and interpreted the results, with support from all other authors. V.T.B. wrote the manuscript with contributions from all other authors.

## Competing interests

The authors declare no competing interests.
