## [Peer Review File · Nature Communications]

Reviewers' comments:

Reviewer #1 (Remarks to the Author):

Review comments:

The manuscript by Bickel et al., presents a new dataset of global distribution of rock falls throughout the manuscript, their spatial variation and their relationships to local geology, in terms of their occurrence on impact craters, tectonic and volcanic features. The authors used lakhs of high-resolution images that cover the entire lunar surface and automatic rock fall detection algorithms and come up with a new dataset that clearly documents the rock fall distribution on the Moon. The dataset shows new knowledge about the lunar rock falls and their spatial association with different geological features of different origins and formation times. I agree with the authors that the manuscript substantially improves our knowledge about the origins of lunar rock falls and recommend that the manuscript could be published by Nature Communications.

The authors correctly relate the formation of boulders to impact cratering processes. In this regard, I would like to point out some more recent publications that may be useful to understand the roles of impact spallation processes on boulder origins. The publications are:

<https://www.sciencedirect.com/science/article/abs/pii/S0019103515004479>

<https://agupubs.onlinelibrary.wiley.com/doi/full/10.1002/2013JE004543>

Formation of boulders is also related to impact fracturing that has been well established on a few terrestrial impact craters. Say, for example:

<https://agupubs.onlinelibrary.wiley.com/doi/10.1029/2005JB003662>

<https://agupubs.onlinelibrary.wiley.com/doi/full/10.1029/2008JE003115>

A few boulder falls have been dated on the Moon. The results can be found in ref [8].

Although I agree that the ages of boulder trails could be less than 20-30 Ma, there is no evidence to suggest that they can be as old as >100 Ma. Hence, the boulder trails are related to erosional processes that occurred in recent rimes but not older than 20-30 Ma. Therefore, the observed rock fall distribution reflects the lunar erosional processes that occurred in the last few million years. I request the authors to add more discussion on the recent geological processes. Of course, the boulders are produced by impacts that occurred over a span of billion years and those sitting on the lunar surface also are also destroyed subsequently. Recent studies also highlighted the survival times of boulders. The authors discussed these aspects but recent processes should be emphasised with more details and clearly link them to the rock falls.

Overall, I enjoyed reading the manuscript. I congratulate the authors for their exciting research work.

My identity may be revealed to the authors:

P. Senthil Kumar

senthil@ngri.res.in

Reviewer #2 (Remarks to the Author):

Comment 1. Line 19-46. It is said in line 19 that rockfall is a process. But in line 46 it is said that authors identified 136,610 rockfalls, probably meaning in this case the traces of this process. It is not clear if authors use term "rockfall" for an individual boulder with visible track or two and more nearby located boulders with tracks are considered as individual rockfall too. It is not clear if a boulder which already lost its track but obviously had it is considered as a rockfall too. This should

be clarified in the very beginning of the paper.

Comment 2. Line 60. This figure shows mean values. Statistical reliability of the conclusion that older lunar regions host boulders with systematically larger volumes should be provided.

Comment 3. Lines 63-67. Figures 2e and 2g shows high concentration of rockfalls associated with Gibbs and S of Rowland R. But why the activity in these locations is considered as recent it is not clear. Some references (to geologic maps) may help and concrete situations should be shown in LROC NAC images as Supplementary figures.

Comment 4. Lines 77-84. Authors write that that most mapped rockfalls are located in craters and that there is no of increase in rockfall density within or in the direct vicinity of tectonic features, such as graben and faults, except several listed cases. They conclude that impacts appear to be the dominant driver of rockfalls while tectonic and volcanism may play some role. But the fact of association of rockfalls with impact craters does not necessarily means that impacts are drivers of this process. Cited in the considered paper work of Kumar et al (2016) describes evidence of very recent tectonic/seismic activity which provoked numerous rockfalls in ancient impact crater Schroedinger. Abundant rockfalls in this crater are due to recent seismicity not to the ancient crater-forming impact. Impact craters are just more numerous on the Moon than volcanic and tectonic features and relatively steep slopes of any origin are the precondition and arena of development of rockfalls mostly caused by shakes of seismic or impact events. So this part of the paper should be reworked.

Reviewer #1

Review comments:

The manuscript by Bickel et al., presents a new dataset of global distribution of rock falls throughout the manuscript, their spatial variation and their relationships to local geology, in terms of their occurrence on impact craters, tectonic and volcanic features. The authors used lakhs of high-resolution images that cover the entire lunar surface and automatic rock fall detection algorithms and come up with a new dataset that clearly documents the rock fall distribution on the Moon. The dataset shows new knowledge about the lunar rock falls and their spatial association with different geological features of different origins and formation times. I agree with the authors that the manuscript substantially improves our knowledge about the origins of lunar rock falls and recommend that the manuscript could be published by Nature Communications.

Our reply: Thank you very much for the recognition and appreciation of our work!

The authors correctly relate the formation of boulders to impact cratering processes. In this regard, I would like to point out some more recent publications that may be useful to understand the roles of impact spallation processes on boulder origins. The publications are:

https://urldefense.proofpoint.com/v2/url?u=https-3A__www.sciencedirect.com_science_article_abs_pii_S0019103515004479&d=DwIF-g&c=vh6FgFnduejNhPPD0fl_yRaSfZy8CWbWnl4XJhSqx8&r=l451AqqPqp7YLqBs9VQoNZ-T0JNKazK8SIN3yJYUNYA&m=Fgld0WMoxd-cGlmINUxzIFRKSPjvuifn2CI8y5hPA3k&s=40TQY-rDRHMzwwiNhZj2OkBpMpdZKVEPF7tptgkZAtU&e=

https://urldefense.proofpoint.com/v2/url?u=https-3A__agupubs.onlinelibrary.wiley.com_doi_full_10.1002_2013JE004543&d=DwIF-g&c=vh6FgFnduejNhPPD0fl_yRaSfZy8CWbWnl4XJhSqx8&r=l451AqqPqp7YLqBs9VQoNZ-T0JNKazK8SIN3yJYUNYA&m=Fgld0WMoxd-cGlmINUxzIFRKSPjvuifn2CI8y5hPA3k&s=1Tlev_9KbSjgxWaYYktCWGzz32caQ0iafDIbdFtJlt4&e=

Formation of boulders is also related to impact fracturing that has been well established on a few terrestrial impact craters. Say, for example:

https://urldefense.proofpoint.com/v2/url?u=https-3A__agupubs.onlinelibrary.wiley.com_doi_10.1029_2005JB003662&d=DwIF-g&c=vh6FgFnduejNhPPD0fl_yRaSfZy8CWbWnl4XJhSqx8&r=l451AqqPqp7YLqBs9VQoNZ-T0JNKazK8SIN3yJYUNYA&m=Fgld0WMoxd-cGlmINUxzIFRKSPjvuifn2CI8y5hPA3k&s=qwNM3rxXBfortfKQVm4GviKK8zflhsi98FhT88YK8il&e=

https://urldefense.proofpoint.com/v2/url?u=https-3A__agupubs.onlinelibrary.wiley.com_doi_full_10.1029_2008JE003115&d=DwIF-g&c=vh6FgFnduejNhPPD0fl_yRaSfZy8CWbWnl4XJhSqx8&r=l451AqqPqp7YLqBs9VQoNZ-T0JNKazK8SIN3yJYUNYA&m=Fgld0WMoxd-cGlmINUxzIFRKSPjvuifn2CI8y5hPA3k&s=hJWgZp68lwr51Bs8F3YItRTeTsO2DhWAoDFB6n4oQs&e=

Our reply: Thank you for suggesting these additional references! As far as we understand, all four references from Dr. Kumar et al. are, to some extent, based on or related to the same fundamental work performed by Polanskey and Ahrens (1990), i.e., artificial impact tests performed in San Marcos gabbro. Thus, citation of all four additional references might be redundant. However, we agree with Dr. Kumar that an additional (terrestrial) reference could be valuable for a reader. Therefore, we added the most relevant (fundamental) of the four references (Kumar, 2005) to the revised manuscript, on page 5.

Added [REF 25, new manuscript]

A few boulder falls have been dated on the Moon. The results can be found in ref [8]. Although I agree that the ages of boulder trails could be less than 20-30 Ma, there is no evidence to suggest that they can be as old as >100 Ma. Hence, the boulder trails are related to erosional processes that occurred in recent rimes but not older than 20-30 Ma. Therefore, the observed rock fall distribution reflects the lunar erosional processes that occurred in the last few million years. I request the authors to add more discussion on the recent geological processes. Of course, the boulders are produced by impacts that occurred over a span of billion years and those sitting on the lunar surface also are also

destroyed subsequently. Recent studies also highlighted the survival times of boulders. The authors discussed these aspects but recent processes should be emphasised with more details and clearly link them to the rock falls.

Our reply: The age estimates derived by Kumar et al. (2019) for one particular lunar crater [REF 8], i.e., ~1.55 and ~1.62 Ma, have been derived with conventional crater counting methods (e.g. Neukum et al., 2001). The track survival times reported in our manuscript have been derived with a combination of (Apollo) in-situ and remote sensing observations and are well established throughout the literature [REFS 2,9,10,11,12]. However, we want to recognize the results of Kumar et al. (2019) and included their minimum survival time estimate in the respective sentences on pages 1, 4, and 5:

“Using these data in combination with orbital imagery, rockfall track survivability has been estimated to range from ~1.55 to 35 Ma^{2,9,10,11,12} with an upper limit of ~150 - 300 Ma¹³, meaning that the tracks observed today are the expression of geologically recent (late Copernican) – and potentially current – erosion processes.”

“Survival times of boulder tracks have been estimated to lie between ~1.55 and 35 Ma using observations made by astronauts, dating of in situ samples, and satellite imagery^{2,9,10,11,12}.”

“Thus, our observations of rockfalls in (pre-)Nectarian terranes (>3.8 Ga old) imply that these ancient terranes have been actively eroded in the last 300 to ~1.55 Ma and likely are subject to erosion today.”

Our reason for assuming maximum track survival times of up to 300 Ma can be found in the Discussion, pages 4 and 5. Based on the assumption that boulders would weather slower than their tracks, we use the estimated survival time of boulders >2m as the upper limit of track survival time, i.e., ~150 to 300 Ma [REF 13]. While this age could potentially be too high, it represents a conservative upper limit.

As Dr. Kumar states, our rockfall distribution map shows a snapshot of the very recent erosion of the lunar surface (in the form of rockfalls). Therefore, all findings in this manuscript are directly or indirectly related to “recent geological processes” of the Moon. For example, on page 3 we describe that “We found evidence that impact processes directly trigger lunar mass wasting (Fig. 2e and g) and, given the likely age of rockfall tracks, actively drive the erosion of the lunar topography (Supplementary Fig. 2). One variation of these impact-ejected boulders features tracks that spread radially away from the impact site (Fig. 2g), where the gradually decreasing distance between individual jumps of each boulder suggests that the boulders were initially airborne and subsequently lost more and more energy along their track. A second variation features an impactor that hits the slope of an existing, larger crater (Fig. 2e), resulting in instant displacement of a number of boulders across the slope of the older host crater. However, the majority of rockfalls occur on crater slopes without the direct influence of a subsequent impactor. This indicates that impacts act not only as a direct trigger, but also induce fracture networks that represent the main preconditioning factor for lunar rockfalls. Subsequently, other, more recent geological processes, such as tectonic and/or volcanic activity, can potentially act as alternative or additional triggers of these impact-fractured rock masses”. We reworked this paragraph and added the last sentence to the manuscript to stress the role of impacts and recent geologic processes in triggering rockfalls.

On pages 3 and 4 we also discuss recent tectonic and volcanic activity that may act as a rockfall trigger. We have updated lines 87 to 95 (new manuscript) to emphasize that this is recent activity, but also to emphasize that with our current data there is some ambiguity: “We do not observe a general increase in rockfall number or spatial density within or in the direct vicinity of tectonic features, such as graben and faults, which would be expected if recent tectonic activity is a main driver of lunar rockfall. Two potential explanations of this observation are: 1) endogenic seismic activity is a relatively unimportant trigger or 2) there is a lack of impact-induced fracture networks (precondition) for high frequency rockfall occurrence in tectonic geomorphic contexts. At present, current data cannot distinguish between these two hypotheses. There are limited exceptions to this observation, such as the rockfall clusters in Vallis Alpes (a graben, Fig. 2a and Supplementary Fig. 3c), Aristarchus plateau (a volcanic region, Supplementary Fig. 3a), and a pyroclastic vent west of the Baily crater (Fig. 2b). Thus, while rockfall frequency and density peak in impact craters, particularly in young and steep craters (e.g. Fig. 3), i.e., lunar rockfalls appear to be predominantly driven by impacts, our results show that recent tectonic and potentially volcanic activity contribute as well. This means that our global map of rockfalls could also be applied to better understand, localize, and constrain the current tectonic – and potentially volcanic – activity of the Moon.”

Another example is supplementary Fig. 3 (new manuscript), which analyses the distribution of rockfalls across different geomorphic and geologic contexts in an attempt to better understand the recent geologic activity and its drivers. We hope that Dr. Kumar will agree with our argumentation and the ample modifications of the text.

Overall, I enjoyed reading the manuscript. I congratulate the authors for their exciting research work.

Our reply: Again, thank you very much for the recognition and appreciation of our work!

My identity may be revealed to the authors:

P. Senthil Kumar

senthil@ngri.res.in

Reviewer #2

Reviewer #2 (Remarks to the Author):

Comment 1. Line 19-46. It is said in line 19 that rockfall is a process. But in line 46 it is said that authors identified 136,610 rockfalls, probably meaning in this case the traces of this process. It is not clear if authors use term "rockfall" for an individual boulder with visible track or two and more nearby located boulders with tracks are considered as individual rockfall too. It is not clear if a boulder which already lost its track but obviously had it is considered as a rockfall too. This should be clarified in the very beginning of the paper.

Our reply: Thank you for pointing out this potential source of confusion. We use the term "rockfall" to describe the characteristic combination of a displaced boulder and its associated track. As suggested, we modified the respective sentence in the introduction to clarify the term "rockfall":

"Rockfalls carve tracks into the lunar surface, which provide a record of the dynamic emplacement process. **The characteristic combination of a displaced boulder or rock fragment and its track – here called rockfall - enables unambiguous detection from satellite imagery (see inset in Fig. 1).**"

Comment 2. Line 60. This figure shows mean values. Statistical reliability of the conclusion that older lunar regions host boulders with systematically larger volumes should be provided.

Our reply: We followed the suggestion and added the number of rockfalls used, the standard distribution, as well as the first and third quartiles of each age class to supplementary figure 1.

Modified supplementary figure 1 & caption

Comment 3. Lines 63-67. Figures 2e and 2g shows high concentration of rockfalls associated with Gibbs and S of Rowland R. But why the activity in these locations is considered as recent it is not clear. Some references (to geologic maps) may help and concrete situations should be shown in LROC NAC images as Supplementary figures.

Our reply: We followed the suggestions and added a new supplementary figure (Fig. 2 in the revised manuscript) that showcases the examples described in the Results section as well as added the following on page 3:

"[...] that impact processes directly trigger lunar mass wasting (Fig. 2e and g) and, **given the likely age of rockfall tracks,** actively drive the erosion of the lunar topography [...]"

Added new supplementary figure 2 & caption

Comment 4. Lines 77-84. Authors write that that most mapped rockfalls are located in craters and that there is no of increase in rockfall density within or in the direct vicinity of tectonic features, such as graben and faults, except several listed cases. They conclude that impacts appear to be the dominant driver of rockfalls while tectonic and volcanism may play some role. But the fact of association of rockfalls with impact craters does not necessarily means that impacts are drivers of this process. Cited in the considered paper work of Kumar et al (2016) describes evidence of very recent tectonic/seismic activity which provoked numerous rockfalls in ancient impact crater Schroedinger. Abundant rockfalls in this crater are due to recent seismicity not to the ancient crater-forming impact. Impact craters are just more numerous on the Moon than volcanic and tectonic features and relatively steep slopes of any origin are the precondition and arena of development of rockfalls mostly caused by shakes of seismic or impact events.

So this part of the paper should be reworked.

Our reply: The term "driver" overarches/connects the two sub-terms "cause" and "trigger" of rockfalls, where causes are processes that lead to rockfalls (e.g. expansion of fracture networks in a rock mass) and triggers are processes that actually trigger a rockfall (e.g. impacts or quakes). The observations of rockfalls that we can make in satellite imagery only in some cases allow for a direct and explicit determination of the cause and/or the trigger, for example, in crater Gibbs and S of Rowland R., where we can directly connect a small, recent impact to rockfall tracks (please see **new supplementary Fig 2.**). Only in these clear cases we directly refer to a "cause" and/or "trigger" (e.g. on page 3). For all other observations, we prefer to use the term "driver", as we cannot distinguish the effective "cause" and "trigger" with a sufficient degree of confidence. We added a new sentence on page 2 that aims to clarify our usage of the term "driver" as well as our reasoning:

"Here, we use the term rockfall "driver" to represent both the long term "cause" (also called precondition) and short term "trigger", as observations in satellite imagery only allow for an explicit distinction of rockfall triggers and causes in some cases^{8,9}."

We also added two references to point a reader to case studies that identified potential local rockfall triggers [REFS 8 and 9] and modified the first sentence of the discussion on page 5 to further clarify the term "driver".

The local observations made by e.g. Kumar et al. (2016) suggest that the rockfalls in Schrödinger basin could have been "triggered" by seismic activity. However, Kumar et al. (2016) themselves state that earthquakes are not the only active trigger in Schrödinger basin: "Therefore, a combination of recent shallow moonquakes and impact events triggered the boulder falls in the Schrödinger basin." We argue that a potential quake was only able to "trigger" rockfalls, because other processes (the "cause" or "causes") created the required predisposition (or precondition), such as impact-induced fracture networks that propagated through the rock mass, producing boulders that can be displaced in the first place. This is why we argue that "impacts drive lunar rockfalls", because they could "cause" and "trigger" rockfalls, or both – we don't know, yet.

In addition, our global observations show that less than 1% of lunar rockfalls are associated with tectonic features. In addition, the spatial density of rockfalls around tectonic features is lower than in craters (please see supplementary Fig. 3). These observations could mean that either 1) seismicity is not the predominant trigger of lunar rockfalls (on a global scale) or that 2) seismicity is not limited to the locations of tectonic features, such as grabens or wrinkle ridges, or a combination of 1) and 2). The results presented in our manuscript substantiate that impacts are the "primary driver" of lunar rockfalls, acting as A) a short term trigger and B) a long term preconditioning factor (cause). We hope that reviewer #2 agrees with our argumentation and with the adapted wording and content.

In response to this comment, as well as a comment from Reviewer #1, we have further updated Lines 72, 74-77, and 87-95 (new manuscript).

Reviewer #3

The paper presents the results of a project aimed at mapping rockfalls on the Moon surface. The result is obtained by using AI techniques applied to high-resolution LROC images from NASA missions.

Results are analysed to understand the geological lunar epoch where rockfalls are located, and to detect the density of these erosion processes on different types of lunar morphology.

Due to the completeness of these outcomes, this study is really important and deserves publication. At the same time, I have some major and minor questions.

Our reply: Thank you very much for the recognition and appreciation of our work!

Major concerns:

In section "Spatial distribution of lunar rockfalls" you mention the examples of a few craters. The criteria for the selection of these craters are not clear: are they well-representative examples? are they only some cases that are well known from the literature? others?

Our reply: We mention these craters by name, because they either feature a distinct asymmetric spatial distribution of rockfalls or particularly high spatial rockfall densities. We modified one of the sentences to clarify the reason we report these craters' names:

"Other remarkable, high spatial density rockfall clusters are located in Mare Orientale basin and the associated Montes Rook and Montes Cordillera, as well as in Tsiolkovsky, Pasteur D, Atlas, Aristoteles, Milne N/L, and Crookes craters."

The authors should discuss more about the precision of the achieved results. Have you validated your result against manually-processed data on small regions?

Our reply: The term precision describes the number of correct rockfall detections in comparison to the number of all (including false) rockfall detections. In other words, precision quantifies (in a percentage) how "useful" and "reliable" the detection results are. For example, a neural network with a precision of 100% would not make any mistakes during classification, i.e., every detection is an actual rockfall and not e.g. a small crater. Following the state-of-the-art of computer vision applications, the precision of the used neural network has been derived for a large number of testing sites using manually (human-) labeled testing data, as described in detail in [REF 20]. The derived precision of the used neural network for a network confidence of 0.6 is 0.98 (98%)²⁰, as reported in the Methods section, pages 8 and 9. In addition, the authors performed an extensive manual study of the produced rockfall distribution map, confirming the previously quantified neural network precision performance.

Small corrections:

- line 36 : "that that" -> "that" (that removed)

- lines 54-57 : this sentence is too long and not too much clear.

Our reply: The four sentences (completely or partially) covered by lines 54 to 57 (old manuscript) are as follows:

"Other remarkable rockfall clusters are located in Mare Orientale basin and the associated Montes Rook and Montes Cordillera, as well as in Tsiolkovsky, Pasteur D, Atlas, Aristoteles, Milne N/L, and Crookes craters."

"The heterogeneous spatial distribution of rockfall clusters could potentially reveal regions of recent seismic activity."

"Lunar rockfall diameters range from ~3 to ~25 m, while the majority of rockfall diameters range from ~7 to ~10 m (Fig. 2)."

"The ubiquitous and homogeneous soil on lunar slopes²¹ would act in favor of larger deposited boulder diameters, since a large portion of the kinetic energy would be absorbed and would, thus, not contribute to boulder fragmentation."

We have carefully re-considered the wording of all four of them and made some minor adjustments to improve their clarity.

- line 74 : you should add here that small craters are neglected due to the fact the CNN applied does not provide good results with small craters, as described in Methods section.

Our reply: The reason for potentially neglecting craters between 1 and 5 km (diameter) is not the performance of our CNN (neural network), which provides similar precision for all used crater diameters. The problem with using the small craters is in the geographic localization accuracy and uncertainties of the available crater maps (vector layers), particularly [REF 22].

The only available global vector map with craters between 1 and 5 km (diameter) is provided by [REF 22] and contains small inaccuracies, like small lateral offsets of the crater polygons from the actual crater, and uncertainties regarding e.g. crater diameters²². For this reason, we report two percentages on page 3, to enable the reader to develop a better understanding of the influence of the used crater diameter range. Craters smaller 1 km cannot be considered at all, because there is no map (vector layer) available. We added a sentence to the Methods, page 10, to highlight the potential inaccuracies associated with [REF 22]:

"It should be noted that some of the used auxiliary data have associated uncertainties and, potentially, errors, such as, for example, the crater map with diameters between 1 and 5 km (diameter)²²."

- line 79 : the expression "E-3" is unusual in a paper, but maybe it's used in Nature Comm.

- Caption of Fig. 1 : it's not clear!

Our reply: We have carefully considered this comment in combination with the following comment and tried to improve the wording and content of Fig. 1's caption. If the reviewer has specific comments for further improving this caption we would be happy to incorporate them.

- Fig. 1 : it's not clear what those spheres represent.

Our reply: The spheres represent the lunar north and south pole in a stereographic projection. We tried to clarify the caption of Fig. 1 by adding the following:

"Heat maps of lunar global rockfall distribution in both equirectangular (70°N to 70°S) and polar projection (top circles, 60°N to 80°N and 60°S to 80°S)."

- line 227 : there is a parenthesis ")" more than necessary. (parenthesis removed)

Our reply: Thank you for these corrections! We applied them throughout the manuscript, where applicable.

REVIEWERS' COMMENTS:

Reviewer #3 (Remarks to the Author):

The authors have appropriately answers to all may major and minor questions. Since my general opinion about the paper was positive and leaning towards publication, now I could firmly recommend to accept th epaper as it is.